# Electromagnetically Induced Transparency-Like Effect by Dark-Dark Mode Coupling

**DOI:** 10.3390/nano11051350

**Published:** 2021-05-20

**Authors:** Qiao Wang, Kaili Kuang, Huixuan Gao, Shuwen Chu, Li Yu, Wei Peng

**Affiliations:** Department of Physics, Dalian University of Technology, Ganjingzi District, Dalian 116024, China; wangqiao@dlut.edu.cn (Q.W.); kellykuang@mail.dlut.edu.cn (K.K.); shark@mail.dlut.edu.cn (H.G.); G21402062@mail.dlut.edu.cn (S.C.); 1050027560@mail.dlut.edu.cn (L.Y.)

**Keywords:** electromagnetically induced transparency, waveguide resonance, surface plasmon polaritons

## Abstract

Electromagnetically induced transparency-like (EIT-like) effect is a promising research area for applications of slow light, sensing and metamaterials. The EIT-like effect is generally formed by the destructive interference of bright-dark mode coupling and bright-bright mode coupling. There are seldom reports about EIT-like effect realized by the coupling of two dark modes. In this paper, we numerically and theoretically demonstrated that the EIT-like effect is achieved through dark-dark mode coupling of two waveguide resonances in a compound nanosystem with metal grating and multilayer structure. If we introduce |1〉, |2〉 and |3〉 to represent the surface plasmon polaritons (SPPs) resonance, waveguide resonance in layer 2, and waveguide resonance in layer 4, the destructive interference occurs between two pathways of |0〉→|1〉→|2〉 and |0〉→|1〉→|2〉→|3〉→|2〉, where |0〉 is the ground state without excitation. Our work will stimulate more studies on EIT-like effect with dark-dark mode coupling in other systems.

## 1. Introduction

The concept of electromagnetically induced transparency (EIT) was first observed in a three-level atomic system. It refers to a quantum destructive interference between two transition processes of atoms tuned by a probe and a driven laser beam, generating a narrow transparency window in a broad opaque spectrum [1]. The EIT is always accompanied by extreme spectral dispersion property, which empowers it with high potential applications in sensitive sensing [2,3], slow light [4,5,6] and metamaterials [7,8]. However, traditional EIT in atoms often requires harsh experimental conditions, such as cryogenic temperature or high intensity laser, thus restricting the practical applications of EIT [9]. By further analyzing the physics behind EIT, researchers found that the EIT can be reproduced through non-quantum approaches, such as using coupled harmonic oscillators [10] or coupled RLC electric circuits [11]. The EIT in non-quantum systems is called electromagnetically induced transparency-like (EIT-like) effect, in analogue to the EIT in atoms. The physical mechanism leads to the realization of EIT-like effect in a number of classical optical systems, for example, waveguide [12,13], photonic crystal structure [14,15], plasmonic nanosystem [16,17,18], and metasurface [19,20], in which the conditions for implementing EIT are not so strict as those in an atomic system. The EIT-like effect is reported for a wide frequency range, from microwave [21], terahertz [22,23], infrared [24,25] to optical frequencies [26]. Recently, the EIT-like effect on metasurfaces has raised intensive attention. Li et al. theoretically and experimentally proposed an actively controlled EIT-like effect in the THz regime on a metasurface using phase change material of vanadium dioxide (VO_2_). The metasurface is composed of meanderline and U-shaped resonators [27]. Xiao et al. designed a graphene-based metasurface, which can exhibit a tunable EIT-like effect at mid-infrared frequency. The EIT-like effect is formed by the coupling of the quadrupole mode of the nano-strip and the dipole mode of the nano-disks [28]. Chen et al. theoretically demonstrated an EIT-like effect at THz frequency on a metasurface, comprising an outer circular ring resonator made by graphene, an inner vertical two-gap circular split ring resonator (VSRR) and a smaller horizontal two-gap circular split ring resonator (HSRR) [29]. Yang et al. proposed a metasurface consisting of a cut-wire and a side coupled split-ring resonator. The metasurface incorporates both EIT-like effect and polarization conversion at THz frequency [30].

Generally, the EIT-like effect in various systems is achieved through (i) bright-dark mode coupling and (ii) bright–bright mode coupling. In the bright-dark mode coupling, the bright mode is a result of direct interaction with the external field and usually processes a low-Q factor. The dark mode does not couple to the external field and is induced by the bright mode through near-field coupling. The dark mode is a high-Q factor resonance. When the bright and dark mode get close to each other at the same resonant frequency, destructive interference between the two modes occurs and an EIT-like effect is generated. Most EIT-like effects reported are of this kind [8,10,16,31,32,33,34]. In the bright-bright mode coupling, the EIT-like effect is realized by breaking the symmetry or varying the lateral distance of two bright modes [35,36,37,38]. Although EIT-like effect has been studied intensively, the EIT-like effect achieved by two dark modes has been seldom reported. Pitchappa et al. reported the active control of EIT-like effect at THz frequency with the excitation of a dual pathway of dark split ring resonators (SRRs) on a metasurface [39]. However, the EIT-like effect with dark-dark mode coupling at visible frequency has not been reported yet.

In this work, we report the EIT-like effect at visible frequency with dark-dark mode coupling of two waveguide resonances in a compound nanosystem. Although we also report the dark-dark mode coupling, our work is different from the work of Pitchappa et al. in structure, frequency and the resonances for coupling. The nanosystem is composed of metal grating and multilayer structure. If we introduce |1〉, |2〉 and |3〉 to represent the surface plasmon polaritons (SPPs) resonance, waveguide resonance in layer 2, and waveguide resonance in layer 4, the destructive interference occurs between two pathways of |0〉→|1〉→|2〉 and |0〉→|1〉→|2〉→|3〉→|2〉, where |0〉 is the ground state without excitation. The formation process of the EIT-like effect is demonstrated by simulated field distributions and theoretical results derived from equations. We further investigate the influences of grating period, grating width, grating thickness, thicknesses of different layers, and refractive indices of different layers on the EIT-like effect. We believe that our work will inspire more studies of EIT-like effect with dark-dark mode coupling in other systems.

## 2. Structure and Theory

The proposed nanosystem is composed of a gold (Au) grating and a multilayer structure with the schematic map shown in Figure 1. The Au grating is defined by its period P, grating width a and thickness h. The multilayer structure contains four layers and a substrate with refractive indices illustrated in Figure 1b. The thicknesses of the four layers are described by $t_{1}$, $t_{2}$, $t_{3}$ and $t_{4}$. We choose a Cytop fluoropolymer material for layer 1, layer 3 and the substrate with a permittivity of $\varepsilon_{1}=\varepsilon_{3}=\varepsilon_{5}=1.8117+2.6900\times 10^{-3}i$ [40]. Additionally, a TiO_2_ material is employed in layer 2 and layer 4 with refractive indices of $n_{2}=n_{4}=2.13$.

The dispersive permittivity of Au is described by a Drude–Lorentz model as [41]: (1)$$\varepsilon_{m}=\varepsilon_{r}-\sum_{j=0}\frac{\omega_{Pj}^{2}}{\omega \left(\omega +i\gamma_{j}\right)}-\sum_{j=0}\frac{\Delta \varepsilon_{j}\Omega_{j}^{2}}{\omega^{2}-\Omega_{j}^{2}+i\omega \Gamma_{j}},$$
where $\varepsilon_{r}$ is the relative permittivity at infinite frequency, $\omega_{Pj}$ refers to the bulk plasma frequency of metal, $\gamma_{j}$ denotes the collision angular frequency, $\Omega_{j}$ represents the Lorentz oscillator strength, $\Gamma_{j}$ stands for the Lorentz spectral width, and $\Delta \varepsilon_{j}$ designates the Lorentz weighting factor. The Drude-Lorentz model takes the interband transitions of metal into consideration and gives a detailed description about Au permittivity for a wide spectral range as experimental data [42].

A two-dimensional finite difference time domain (FDTD) method is employed in the numerical study [43]. The FDTD method was first proposed by Yee in 1966. The method divides the computational space into many cubic Yee cells. In every Yee cell, each electric (magnetic) field component is surrounded by four magnetic (electric) field components. The spatial setting of the electromagnetic field satisfies Faraday’s law and Ampere’s law. The method replaces the partial differential format of Maxwell’s equations with the difference of electromagnetic field components of adjacent cells. The electromagnetic field distribution is obtained over the entire space by solving Maxwell’s equations with a gradual increase in time step. This method is more suitable for dealing with electromagnetic scattering and radiation of a complex target and is therefore competent for the study of the proposed nanosystem. The mesh size is 2 nm×2 nm in the simulated region, which is sufficiently fine for describing the nanosystem. The periodic and perfect match layer boundaries are applied on *x* and *y* directions, respectively. A *p*-polarized plane wave with a wavenumber of k enters from the grating side. 

There are two types of resonances in the proposed nanosystem: SPPs resonance and waveguide resonance. SPPs are the collective oscillation of free electrons at the Au/dielectric interface, and the wavenumber is expressed by [44]:(2)$$k_{SPP}=k\sqrt{\frac{\varepsilon_{m}{}^{\prime }\varepsilon_{d}}{\varepsilon_{m}{}^{\prime }+\varepsilon_{d}}},$$
where k and $k_{SPP}$ are the wavenumbers of incident wave and SPPs, respectively, $\varepsilon_{m}{}^{\prime }$ is the real part of Au permittivity, and $\varepsilon_{d}$ is the permittivity of dielectric close to the metal. The SPPs resonance is excited by a grating when the propagating constant of the grating equals the wavenumber of SPPs, i.e., $\beta =k_{SPP}$. The propagating constant of the grating is written as:(3)β=ksinθ±νG,
where θ is the incident angle, ν is a positive integer of 1, 2, 3 … and G=2π/P represents the lattice vector of the grating.

In the multilayer structure, two waveguides are formed in layer 2 and layer 4. The waveguide resonances in layer 2 and layer 4 are TM components because the electric field of the incident wave lays in the *xy* plane. The characteristic equations of the TM components of waveguide resonances in layer 2 and layer 4 are written as Equations (4) and (5), respectively [45]:(4)$$\kappa_{2}d_{2}=m\pi +\arctan\left(\frac{n_{2}^{2}\kappa_{1}}{n_{1}^{2}\kappa_{2}}\right)+\arctan\left(\frac{n_{2}^{2}\kappa_{3}}{n_{3}^{2}\kappa_{2}}\right),$$
(5)$$\kappa_{4}d_{4}=n\pi +\arctan\left(\frac{n_{4}^{2}\kappa_{3}}{n_{3}^{2}\kappa_{4}}\right)+\arctan\left(\frac{n_{4}^{2}\kappa_{5}}{n_{5}^{2}\kappa_{4}}\right),$$
where $\kappa_{1}=\sqrt{{\beta^{\prime }}^{2}-n_{1}^{2}k_{0}^{2}}$, $\kappa_{3}=\sqrt{{\beta^{\prime }}^{2}-n_{3}^{2}k_{0}^{2}}$, $\kappa_{5}=\sqrt{{\beta^{\prime }}^{2}-n_{5}^{2}k_{0}^{2}}$, $\kappa_{2}=\sqrt{n_{2}^{2}k_{0}^{2}-{\beta^{\prime }}^{2}}$ and $\kappa_{4}=\sqrt{n_{4}^{2}k_{0}^{2}-{\beta^{\prime }}^{2}}$. The $\beta^{\prime }$ and $k_{0}$ represent the propagating constant and wavenumber of waveguide resonance, respectively. The variable m or n of 0, 1, 2… corresponds to waveguide resonances of ${\mathrm{TM}}_{0}$, ${\mathrm{TM}}_{1}$, ${\mathrm{TM}}_{2}$ … in layer 2 or layer 4, respectively. When the propagating constant of the waveguide equals that of grating, that is: (6)$$\beta^{\prime }=\beta ,$$
the waveguide resonances are excited by the SPPs. Using simultaneous Equations (3)–(6), one can derive the theoretical $k_{0}$ of the waveguide resonances in layer 2 and layer 4. The corresponding theoretical wavelengths of the waveguide resonances are given out by $2\pi /k_{0}$.

## 3. Results and Discussions

We start with reflection spectra of different thicknesses of layer 1, i.e., $t_{1}$, as shown in Figure 2a. The parameters of the grating are set as $P=500_{}\;\mathrm{nm}$, $a=400_{}\;\mathrm{nm}$ and $h=50_{}\;\mathrm{nm}$. The thicknesses of layer 2, layer 3 and layer 4 are $t_{2}=100_{}\;\mathrm{nm}$, $t_{3}=640_{}\;\mathrm{nm}$ and $t_{4}=100_{}\;\mathrm{nm}$. The incident angle of θ is zero, indicating a normal incidence considered. Figure 1a shows the reflection spectra with different $t_{1}$. It is found that the EIT-like effect gradually disappears as $t_{1}$ increases. By considering the symmetry and contrast of the result, we choose $t_{1}=300_{}\;\mathrm{nm}$ to illustrate the EIT-like effect in Figure 2b. The reflections with $t_{1}=1200_{}\;\mathrm{nm}$ are also displayed as a comparison. The corresponding positions of these two parameters are also indicated as dashed lines in Figure 2a. Figure 2b clearly exhibits the EIT-like effect for $t_{1}=300_{}\;\mathrm{nm}$ in the red curve. The peak and dips of the EIT-like effect are marked as point A, B and C for simplicity. To reveal the physical mechanism, we analyze the EIT-like effect from two respects: detailed field distributions and theoretical results of equations. Figure 3 gives out the detailed magnetic field distributions of points A, B and C. The field distributions clarify that the EIT-like effect is formed by the destructive interference of the two waveguide resonances in layer 2 and layer 4. The peak of the EIT-like effect is induced by the waveguide resonance in layer 4 in Figure 3b. The simulated wavelength of the peak of EIT-like effect (i.e., point B in the red curve) is 720.8 nm. The theoretical wavelength of the peak of EIT-like effect is 720.0 nm, which is derived from the simultaneous Equations (3), (5) and (6) elaborated in the Structure and Theory section. The simulated wavelength of the peak of EIT-like effect is consistent with the theoretical wavelength (marked by the red line on the top of Figure 2b), demonstrating the origin of the EIT-like effect. For $t_{1}=1200_{}\;\mathrm{nm}$, there is no EIT-like effect, as shown in the black curve in Figure 1b, because the bright SPPs mode is too far to excite the waveguide resonances in layer 2 and layer 4. This proves that the EIT-like effect is a result of the coupling of two dark waveguide modes.

Then, we make an analogy of the EIT-like effect to the transition processes of an atom. The nanosystem without excitation is compared with the ground state of |0〉 in an atom. The SPPs resonance, waveguide resonance in layer 2 and waveguide resonance in layer 4 are represented by the excited states of |1〉, |2〉 and |3〉. When the propagating constant of the grating equals the wavenumber of SPPs, the SPPs resonance is excited by the incident wave, which corresponds to the transition process of |0〉→|1〉. If the thickness of $t_{1}$ is proper, the waveguide resonance in layer 2 is excited by SPPs resonance, i.e., the process of |1〉→|2〉. Therefore, the pathway of exciting waveguide resonance in layer 2 is |0〉→|1〉→|2〉. With proper $t_{3}$, the waveguide resonance in layer 2 further excites waveguide resonance in layer 4 (this will be further explained in Figure 4b). Therefore, the pathway of exciting waveguide resonance in layer 4 is |0〉→|1〉→|2〉→|3〉. To sum up, the EIT-like effect is formed by the destructive interference of the two pathways of |0〉→|1〉→|2〉 and |0〉→|1〉→|2〉→|3〉→|2〉.

Subsequently, the influences of $t_{2}$, $t_{3}$ and $t_{4}$ on the EIT-like effect are investigated, respectively, as shown in Figure 4. The basic parameters of $t_{2}$, $t_{3}$ and $t_{4}$ are 100 nm, 640 nm and 100 nm. When one of the three changes, the other two remain unchanged. The other parameters are set as $P=500_{}\;\mathrm{nm}$, $a=400_{}\;\mathrm{nm}$, $h=50_{}\;\mathrm{nm}$ and $t_{1}=300_{}\;\mathrm{nm}$. Figure 4a shows that the EIT-like effect appears repeatedly as $t_{2}$ increases due to higher orders of waveguide resonance in layer 2. The theoretical $t_{2}$ for higher orders of waveguide resonance of ${\mathrm{TM}}_{0}$, ${\mathrm{TM}}_{1}$ and ${\mathrm{TM}}_{2}$ are 100 nm, 329 nm and 559 nm, which are derived from Equations (3), (4) and (6). The positions of $t_{2}$ of 100 nm, 329 nm and 559 nm are marked by black dashed lines in Figure 4a, which correspond to the calculated results. The dips of the EIT-like effect at $t_{2}=329_{}\;\mathrm{nm}$ and $t_{2}=559_{}\;\mathrm{nm}$ are marked as points D and E. The magnetic field distributions of points D and E are illustrated, respectively, in Figure 5a,b, which exhibit two maximums for ${\mathrm{TM}}_{1}$ and three maximums for ${\mathrm{TM}}_{2}$ along the *y* direction in layer 2. The magnetic field distribution of the dip of the EIT-like effect for ${\mathrm{TM}}_{0}$ at $t_{2}=100_{}\;\mathrm{nm}$ has already been shown in Figure 3a. Through the above analysis, the blue bands in Figure 4a stand for different orders of waveguide resonance in layer 2 and the unchanged narrow red line at 720.8 nm represents waveguide resonance in layer 4.

Figure 4b clearly demonstrates that the thickness of layer 3 (i.e., $t_{3}$) determines the bandwidth of the peak of EIT-like effect. As $t_{3}$ increases, the bandwidth of the peak decreases gradually. When $t_{3}>900_{}\;\mathrm{nm}$, the EIT-like effect almost disappears because the two waveguide resonances cannot couple to each other for such a large distance between them. Therefore, only waveguide resonance in layer 2 remains in the spectrum. The explanation is supported by testing the field distribution at the dip of the reflection spectrum for $t_{3}>900_{}\;\mathrm{nm}$ (not shown in the paper). 

Figure 4c shows the reflection spectra with different $t_{4}$. Similarly, the EIT-like effect also appears repeatedly as $t_{4}$ increases because higher orders of waveguide resonance emerge in layer 4. Since we did not change $t_{2}$, the wide blue line represents the unchanged waveguide resonance in layer 2. The theoretical $t_{4}$ for ${\mathrm{TM}}_{0}$, ${\mathrm{TM}}_{1}$ and ${\mathrm{TM}}_{2}$ are the same as that of $t_{2}$, i.e., 100 nm, 329 nm and 559 nm. The peaks of the EIT-like effect at $t_{4}=329_{}\;\mathrm{nm}$ and $t_{4}=559_{}\;\mathrm{nm}$ are marked as points F and G in Figure 4c. Figure 5c,d exhibit the magnetic field distributions of points F and G with ${\mathrm{TM}}_{1}$ and ${\mathrm{TM}}_{2}$ in layer 4.

In the previous discussions, we set $t_{2}=t_{4}=100\;\mathrm{nm}$ for simplicity. Actually, the EIT-like effect is formed with $t_{2}=t_{4}$ for the same order as the waveguide modes in layer 2 and layer 4. We have also tested the reflection spectra of $t_{2}=t_{4}=90_{}\;\mathrm{nm}$ and $t_{2}=t_{4}=110_{}\;\mathrm{nm}$. There is an EIT-like effect appearing in these two cases because resonant wavelengths of waveguide modes in layer 2 and layer 4 shift towards the same direction in the spectrum according to Equations (4) and (5). $t_{3}$ determines the bandwidth of the peak of EIT-like effect, so the choice of $t_{3}$ is not very strict. We set $t_{3}=640_{}\;\mathrm{nm}$ by considering the bandwidth and contrast of the EIT-like effect. 

Next, we change the refractive indices of layer 1, layer 3 and the substrate. The structural parameters are $P=500_{}\;\mathrm{nm}$, $a=400_{}\;\mathrm{nm}$, $h=50_{}\;\mathrm{nm}$, $t_{1}=300_{}\;\mathrm{nm}$, $t_{2}=100_{}\;\mathrm{nm}$, $t_{3}=640_{}\;\mathrm{nm}$ and $t_{4}=100_{}\;\mathrm{nm}$. Figure 6a–c show the reflection spectra with different $n_{1}$, $n_{3}$ and $n_{5}$, respectively. It can be derived from Equations (4) and (5) that the increasing refractive index of the material next to the waveguide will lead to a redshift in the wavelength of the waveguide resonance. The simulation results in Figure 6a–c show the same redshift trend as $n_{1}$, $n_{3}$ or $n_{5}$ increases. However, the phenomena are different. Particularly, the waveguide resonance in layer 2 redshifts with an increasing $n_{1}$ because layer 1 is adjacent to layer 2. Therefore, a redshifted waveguide resonance in layer 2 and an unchanged waveguide resonance in layer 4 form a Fano shape on the left falling edge in Figure 6a. Similarly, the waveguide resonance in layer 4 redshifts with an increasing $n_{5}$ because the substrate is next to layer 4. Consequently, a redshifted waveguide resonance in layer 4 and an unchanged waveguide resonance in layer 2 result in a Fano resonance on the right falling edge in Figure 6c. If we increase $n_{3}$, the two waveguide resonances in layer 2 and layer 4 both redshift. Hence, the EIT-like effect still appears at a longer wavelength in Figure 6b. The schematic representations of the formation processes of Fano resonance and EIT-like effect with the increasing refractive indices are shown in the insets of Figure 6a–c.

Afterwards, the influences of $n_{2}$ and $n_{4}$ on the EIT-like effect are investigated, as shown in Figure 7a,b. The refractive index of 2.13 denotes the TiO_2_ material employed in layer 2 and layer 4 in the previous discussions. Additionally, the refractive indices of 2.00 and 2.198 represent the material of Si_3_N_4_ and Ge-doped SiO_2_, respectively. Whether we increase $n_{2}$ or $n_{4}$, the wavelength of the waveguide resonance in layer 2 or layer 4 redshifts according to Equations (4) and (5). The waveguide resonance in layer 2 redshifts with $n_{2}=2.198$ compared with that of $n_{2}=2.13$. Therefore, a redshifted waveguide resonance in layer 2 and an unchanged waveguide resonance in layer 4 form a Fano resonance on the left falling edge, as seen in the blue curve in Figure 7a. For $n_{4}=2.198$, the waveguide resonance in layer 4 redshifts compared with that of $n_{4}=2.13$. A redshifted waveguide resonance in layer 4 and an unchanged waveguide resonance in layer 2 form a Fano resonance on the right falling edge, as seen in the blue curve in Figure 7b. Through the same way of analyzing, the simulated results with $n_{2}=2.00$ and $n_{4}=2.00$ can also be understood.

We further examine the grating parameters on the EIT-like effect. Figure 8a,b show the reflection spectra with different $t_{1}$ for $P=600_{}\;\mathrm{nm}$ and $P=700_{}\;\mathrm{nm}$. The other parameters are $a=400_{}\;\mathrm{nm}$, $h=50_{}\;\mathrm{nm}$, $t_{2}=100_{}\;\mathrm{nm}$, $t_{3}=640_{}\;\mathrm{nm}$ and $t_{4}=100_{}\;\mathrm{nm}$. Compared with $P=500_{}\;\mathrm{nm}$ in Figure 1a, the EIT-like effect redshifts with an increasing P. The reason of the redshift of the EIT-like effect is explained as follows. When the grating period of P increases, the wavenumber of the SPPs resonance decreases according to Equations (2) and (3). The SPPs resonance excites the waveguide resonance, so the propagating constant $\beta^{\prime }$ of the waveguide resonance decreases. Through Equations (4) and (5), one can derive that the wavelengths of the two waveguide resonances redshift. Figure 8c illustrates the reflection spectra with a continuous change of P for $t_{1}=300_{}\;\mathrm{nm}$. It indicates that the peak of the EIT-like effect is broadening as P increases.

Figure 9a,b show reflection spectra with different grating widths a and thicknesses h. The other parameters are $P=500_{}\;\mathrm{nm}$, $t_{1}=300_{}\;\mathrm{nm}$, $t_{2}=100_{}\;\mathrm{nm}$, $t_{3}=640_{}\;\mathrm{nm}$ and $t_{4}=100_{}\;\mathrm{nm}$. Figure 9a exhibits that the EIT-like effect is not explicit for a small a. This is due to the fact that the grating width is too small to sustain SPPs resonance. Therefore, the corresponding waveguide resonances in layer 2 and layer 4 are not excited. Figure 9b illustrates the evolution of the two waveguide resonances with an increasing h. For comparison, we give out the reflection spectra with different h of a nanosystem with Au grating and a fluoropolymer substrate in Figure 9c. As h increases, only 0th and 1st orders of Fabry-Pérot (FP) resonance in the grating slit are indicated. In Figure 9b, the reflection spectra become complicated for 0th and 1st orders of FP resonance in the grating couple with the two waveguide resonances.

Finally, we make a summary of the EIT-like effect achieved in our proposed nanosystem and in other different nanosystems in Table 1. Zhang’s group introduced an EIT-like effect in plasmonic metamaterial by the coupling of bright dipole antenna and dark mode consisting of two parallel metal strips with a small separation [34]. Hu et al. reported an EIT-like effect with a bar resonator and two split SRRs [46]. Taubert et al. theoretically and experimentally demonstrated an EIT-like effect by the coupling of a broad bright dipolar to a narrow dark quadrupolar plasmon resonance [10]. Luo et al. achieved EIT-like effect in mid-infrared based on the strong coupling between localized and propagating plasmonic modes in layered graphene ribbon-grating and continuous sheet systems [47]. Vafapour et al. exhibited the EIT-like effect with two vertical bars as a bright plasmonic mode and a horizontal bar as a dark plasmonic mode [48]. Wei et al. reported the EIT-like effect due to the destructive interference between the bright and dark resonator of a dielectric metasurface consisting of two mutually perpendicular silicon-based nanoscale bars [49]. Zhang et al. presented the EIT-like effect in an MIM waveguide with a stub coupled ring resonator [50]. Wei et al. reported a tunable EIT-like effect in a graphene-based silicon–air grating structure [51]. He et al. realized the EIT-like effect based on the bright–bright mode coupling in a stacked metal-dielectric MM in the near-infrared regime [37]. Jin et al. studied the EIT-like effect at optical frequencies in a metasurface composed of two silver strips as two bright modes [52]. Fu et al. realized a tunable EIT-like effect in the mid-infrared region by using two parallel graphene nanostrips [38]. Our work is different from these previous reported nanosystems. The EIT-like effect in our proposed nanosystem is generated by two dark waveguide modes coupling at a visible regime.

## 4. Conclusions

In summary, we report an EIT-like effect with the coupling of two dark waveguide modes in the proposed nanosystem. The excitation of the EIT-like effect is demonstrated by simulated field distributions and theoretical equations. As the thickness of layer 2 or layer 4 increases, the EIT-like effect appears repeatedly due to the appearance of high orders of the TM component, i.e., ${\mathrm{TM}}_{1}$ and ${\mathrm{TM}}_{2}$. The thickness of layer 3 determines the bandwidth of the peak of the EIT-like effect. An increase in any refractive index among $n_{1}$, $n_{2}$, $n_{4}$ and $n_{5}$ will result in a Fano resonance because of the coupling of a redshifted waveguide resonance and an unchanged waveguide resonance. When the refractive index of $n_{3}$ increases, the EIT-like effect still appears at a longer wavelength because both waveguide resonances redshift. We further examine the influence of grating parameters on the EIT-like effect. As the grating period of P increases, the EIT-like effect redshifts and the bandwidth of its peak broadens. The EIT-like is not explicit for a small a due to the fact that the grating width is too small to sustain SPPs. As the grating thickness of h increases, the reflection spectra become complicated because the two waveguide resonances couple with 0th and 1st orders of FP resonance in the grating slit. Our work might stimulate more research on the EIT-like effect with two dark mode coupling.

## Figures and Tables

**Figure 1 nanomaterials-11-01350-f001:**
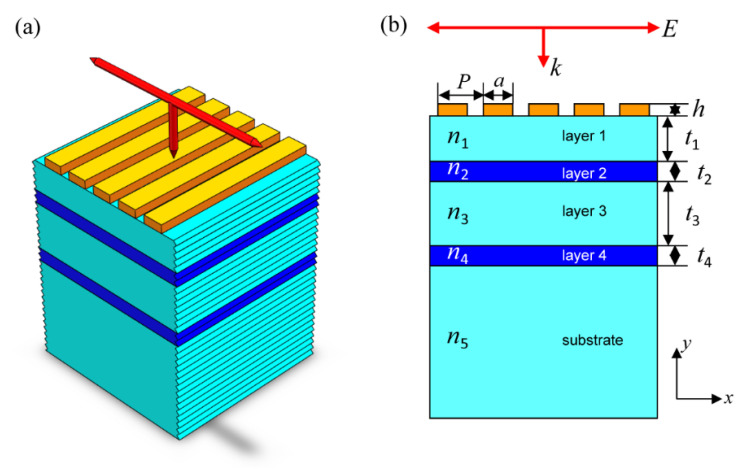
Schematic map of the proposed nanosystem: (**a**) Main view; (**b**) Side view.

**Figure 2 nanomaterials-11-01350-f002:**
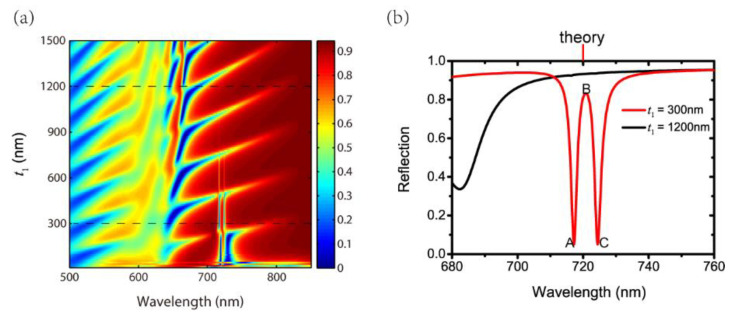
(**a**) Reflection spectra with different $t_{1}$; (**b**) Reflections with $t_{1}=300_{}\mathrm{nm}$ and $t_{1}=1200_{}\mathrm{nm}$.

**Figure 3 nanomaterials-11-01350-f003:**
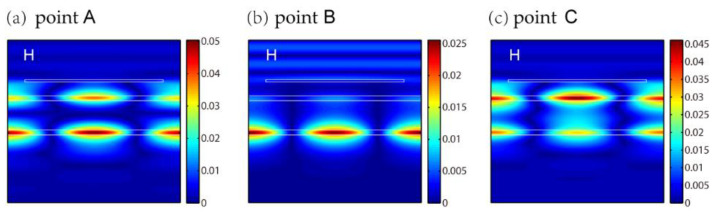
Magnetic field distributions of points (**a**) A, (**b**) B and (**c**) C in Figure 2b.

**Figure 4 nanomaterials-11-01350-f004:**
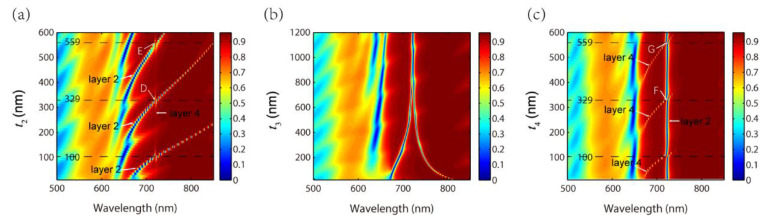
Reflection spectra with different (**a**) $t_{2}$, (**b**) $t_{3}$ and (**c**) $t_{4}$.

**Figure 5 nanomaterials-11-01350-f005:**
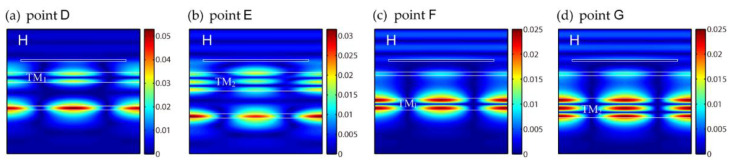
Magnetic field distributions of points (**a**) D, (**b**) E, (**c**) F and (**d**) G in Figure 4a,c.

**Figure 6 nanomaterials-11-01350-f006:**
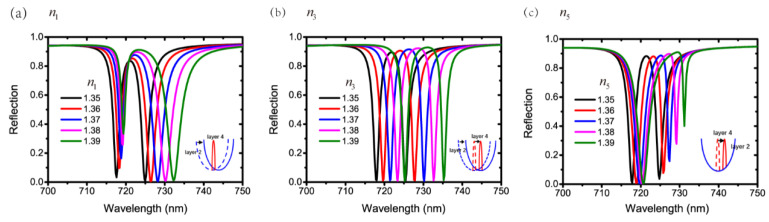
Reflection spectra with different (**a**) $n_{1}$, (**b**) $n_{3}$ and (**c**) $n_{5}$.

**Figure 7 nanomaterials-11-01350-f007:**
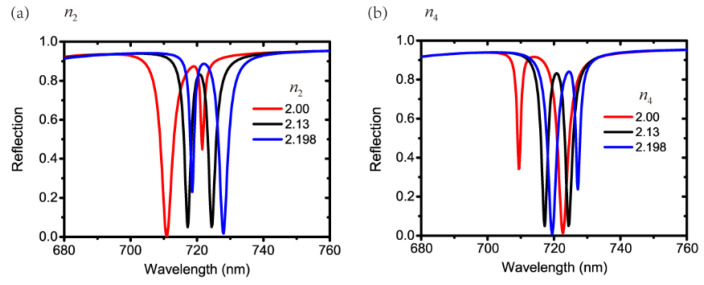
Reflection spectra with different (**a**) $n_{2}$ and (**b**) $n_{4}$.

**Figure 8 nanomaterials-11-01350-f008:**
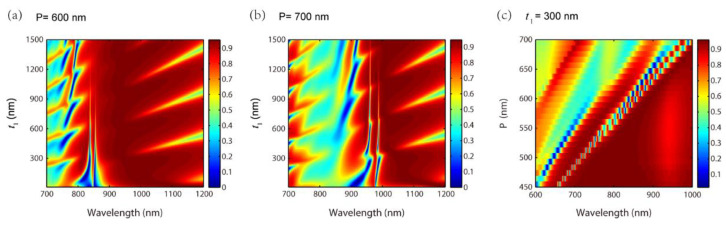
Reflection spectra with different $t_{1}$ for (**a**) $P=600_{}\mathrm{nm}$ and (**b**) $P=700_{}\mathrm{nm}$; (**c**) Reflection spectra with different P for $t_{1}=300_{}\mathrm{nm}$.

**Figure 9 nanomaterials-11-01350-f009:**
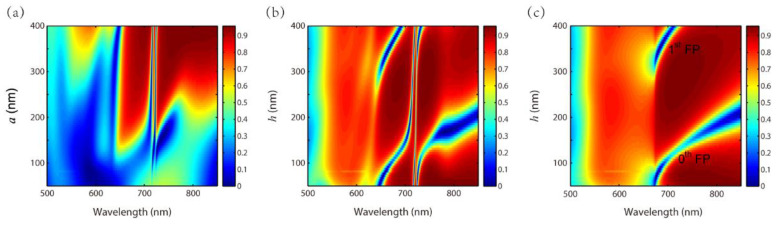
Reflection spectra with different (**a**) a and (**b**) h of the proposed nanosystem; (**c**) Reflection spectra with different h of a nanosystem with Au grating and a fluoropolymer substrate.

**Table 1 nanomaterials-11-01350-t001:** The summary of EIT-like effect in different nanosystems.

Structure	Type	Coupling Manner	Frequency
Ag dipole antenna and two parallel Ag strips [34]	metasurface	bright-dark	near-infrared
One bar resonator and two split ring resonators [46]	metasurface	bright-dark	near-infrared
Au nano-cut-wire and quadrupole wire pair [10]	metamaterial	bright-dark	near-infrared
Graphene ribbon-grating/dielectric layer/graphene sheet [47]	metamaterial	bright-dark	mid-infrared
Three Ag bars [48]	metasurface	bright-dark	near-infrared
Two perpendicular Si-based nanoscale bars [49]	metasurface	bright–dark	near-infrared
MIM waveguide with stub coupled ring resonator [50]	metasurface	not mentioned	near-infrared
Graphene sheet/silicon–air grating/graphene sheet [51]	metamaterial	not mentioned	mid-infrared
Ag strip/SiO_2_ layer/Si rod [37]	metamaterial	bright-bright	near-infrared
Two Ag strips [52]	metasurface	bright-bright	visible
Two graphene strips [38]	metasurface	bright-bright	mid-infrared
Au grating and a multilayer structure	metamaterial	dark-dark	visible

## Data Availability

The data presented in this study are available on request from the corresponding author.

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
