# Peer review of "Electromagnetically Induced Transparency-Like Effect by Dark-Dark Mode Coupling"

_nanomaterials, 2021, doi:10.3390/nano11051350_

Round 1

Reviewer 1 Report

The authors in this theoretical work present electromagnetically induced transparency-like (EIT-like) effect at the visible frequency with dark-dark mode coupling of two waveguide resonances in a compound nanosystem of metal grating and multilayer structure. The paper is well written and the results are interesting, however, the disadvantage of this work is the lack of an experiment. Nevertheless, this work can be accepted for publication.

Author Response

Dear reviewer,

Dear reviewer, We would like to express our sincere thanks to your positive and valuable comments on our manuscript. The detail response please see the attachment.

Reviewer 2 Report

The paper is devoted to investigation of the mode coupling in the nanostructure that consists of the two planar waveguides and the metal diffraction grating. The surface plasmon polaritons resonance, waveguide resonances in both planar waveguides are considered as the excited states in 3-level atom.

This analogy allowed the authors to predict an interference effect which is similar to the electromagnetically induced transparency effect. This effect was named EIT-like effect . The main part of the article contains detailed description of the numerical simulation results.

 The paper contains results that may be representing interest for experts in area of the nanooptics. I believe the manuscript is suitable for publication in the Nanomatetials.

Author Response

Dear reviewer,

We would like to express our sincere thanks to your positive and valuable comments on our manuscript. The detail response please see the attachment.

Reviewer 3 Report

The manuscript entitled "Electromagnetically Induced Transparency-like Effect by Dark-Dark Mode Coupling"  presents an interesting approach and by using multi-layer structure comprising Au grating, Cytop fluoropolymer and TiO2, proves that the EIT-like effect is a result of the coupling of two dark wave-guide modes.

Nevertheless, some aspects need to be clarified/completed or corrected:

1. Punctuation, spaces require revision.

2. References are needed at the end of this phrase, that needs also reformulation: ''Although EIT-like effect has been studied intensively, the EIT-like effect achieved by two dark modes has been seldom reported [REFERENCES XXXXX]. While the EIT-like effect with dark-dark mode coupling is also an important category for a well established EIT theory."

3. Please thoroughly discuss the importance of the thickness for the process and the reason why the Authors selected that values, as you did for the grating parameters.

4. Recent references in the field (2020/2021) should complete the paper and they can be comparatively discussed with your results.

5. A discussion regarding other system of nanomaterials is requested maybe insert a Table).

Author Response

(The authors gave the same response as above.)
